# Cycle-CTFlow: A CycleGAN–Normalizing Flow Harmonization Framework for Improved Nodule Detection

**Anil Yadav**[*1,2] (iD)                                           AYADAV01@UCLA.EDU

[1] *Department of Bioengineering, Samueli School of Engineering, University of California, Los Angeles, 90095, CA, USA*

[2] *Medical & Imaging Informatics Group, Department of Radiological Sciences, David Geffen School of Medicine, UCLA, Los Angeles, 90095, CA, USA*

**Nathan Tran**[2]                                           NATHANTRAN3819@GMAIL.COM

**Spencer Welland**[3]                                        SWELLAND@MEDNET.UCLA.EDU

[3] *Center for Computer Vision and Imaging Biomarkers, Department of Radiological Sciences, David Geffen School of Medicine at UCLA, Los Angeles, CA 90095, USA*

**John M. Hoffman**[3]                                       JMHOFFMAN@MEDNET.UCLA.EDU

**Grace Hyun J. Kim**[3]                                      GRACEKIM@MEDNET.UCLA.EDU

**Ashley E. Prosper**[2]                                      APROSPER@MEDNET.UCLA.EDU

**Denise R. Aberle**[1,2]                                     DABERLE@MEDNET.UCLA.EDU

**Michael F. McNitt-Gray**[3]                                MMCNITTGRAY@MEDNET.UCLA.EDU

**William Hsu**[1,2]                                          WHSU@MEDNET.UCLA.EDU

**Editors:** Accepted for publication at MIDL 2025

## Abstract

While a number of artificial intelligence (AI) algorithms exist for diagnostic tasks like nodule detection and classification, their clinical adoption is hindered by significant variability in computed tomography (CT) acquisitions, which impacts AI performance. Variations in scanner hardware, reconstruction methods, dose levels, and patient demographics cause domain shift, underscoring the need for harmonization and adaptation strategies to ensure consistent performance across diverse settings. To address this challenge, we propose a method that leverages knowledge of the AI model's training distribution to guide the training of a normalizing flow-based harmonization method. The goal is to harmonize input images so that their distribution closely aligns with that of the downstream model's training domain. Our proposed approach, Cycle-CTFlow, leads to improvements in nodule detection performance: a 5.2% increase in sensitivity and a 2.6% increase in competition performance metric (CPM) compared to no harmonization on the MiniDeepLesion dataset, and a 1.9% increase in sensitivity and an 8.5% increase in CPM on the UCLA in-house diagnostic dose dataset.

**Keywords:** Computed Tomography, Harmonization, Normalizing Flow

## 1. Introduction

Computed tomography (CT) plays a pivotal role in the detection and management of various diseases, such as lung cancer, where early and accurate identification of pulmonary nodules

---

[*] Corresponding author

can significantly improve patient outcomes (National Lung Screening Trial Research Team et al., 2011). The rapid growth of CT datasets has fueled a surge in AI-driven diagnostic tools (Hosny et al., 2018); however, deploying these models in real-world clinical settings remains challenging due to limited generalizability. Substantial variability in CT images—due to differences in scanner hardware, acquisition settings, and patient populations—can lead to domain shifts that degrade the performance of deep learning models when applied to unseen data (Yadav et al., 2025).

We propose a generative adversarial network paired with a normalizing flow-based model to align the harmonization output with the target distribution of the downstream task. We select an open-source MONAI nodule detection model trained on the Lung Nodule Analysis 2016 (LUNA16) dataset and aim to improve its generalizability to external datasets. We train a normalizing flow model on our in-house low-dose CT (LDCT) dataset with paired high-dose scans, and introduce a CycleGAN that transforms high-dose images to match LUNA16 texture. We hypothesize that mapping LDCT images to these texture-aligned targets will enhance downstream detection performance (i.e., nodule detection).

## 2. Methods

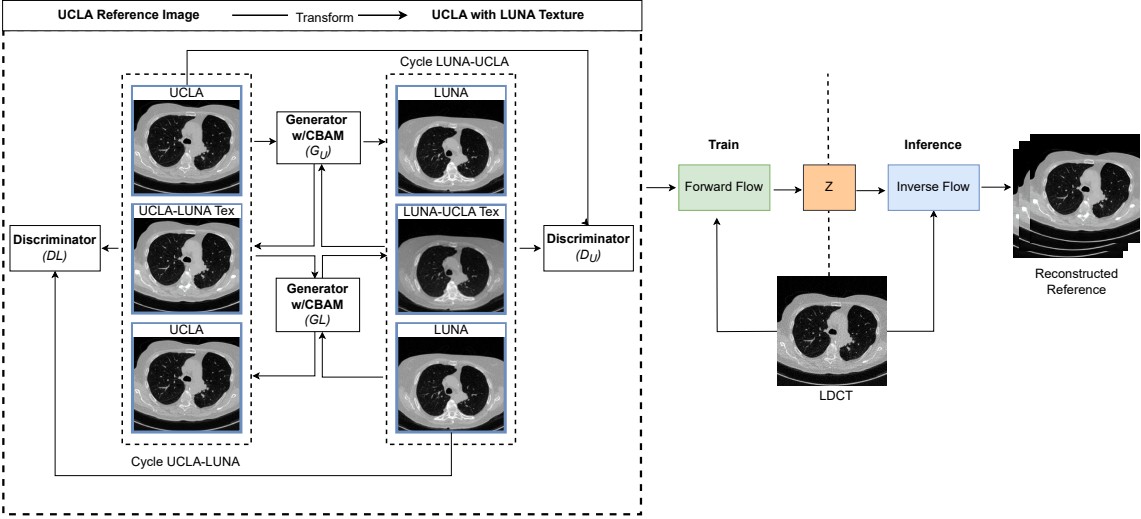

Figure 1: Training scheme of the Cycle-CTFlow model.

We used two datasets for training. The first was an in-house dataset of 100 LDCT exams (31,308 slices), acquired on a Siemens Definition AS64 scanner with 1mm slices using the B45 kernel (i.e., reference condition). To simulate a lower-dose setting, Poisson noise equivalent to 25% of the original dose was added to the raw projection data—referred to as the Medium/25% condition. The second dataset was LUNA16, from which we extracted an equal number of slices to ensure consistency across domains.

We first trained a CBAM-integrated CycleGAN (You et al., 2019) on 30% of 31,308 slices to transform the texture of reference condition images to the LUNA16 domain. Prior work has shown that this integration improves PSNR and SSIM over the standard CycleGAN. On

Table 1: Performance comparison of harmonization methods across two datasets. Competition performance metric (CPM) represents the average sensitivity at seven predefined false positive rates, as defined by the LUNA16 evaluation criteria.

| Dataset | Model | TP | FP | FN | Sensitivity | CPM |
|---|---|---|---|---|---|---|
| MiniDeepLesion | No Harmonization | 76 | 442 | 12 | 0.8636 | 0.5633 |
| | CTFlow-Base | 77 | 1447 | 11 | 0.8750 | 0.5649 |
| | Cycle-CTFlow | 80 | 454 | 8 | 0.9090 | 0.5779 |
| UCLA in-house | No Harmonization | 104 | 445 | 8 | 0.9285 | 0.6011 |
| | CTFlow-Base | 103 | 509 | 9 | 0.9196 | 0.6172 |
| | Cycle-CTFlow | 106 | 534 | 6 | 0.9464 | 0.6525 |

10% held-out data, this model achieved a mean PSNR of 30.908 and SSIM of 0.875. Next, we trained a normalizing flow model, CTFlow (Wei et al., 2023), on 70% of the data. It was previously shown to outperform baseline generative models in handling CT reconstruction variability. In this model, the latent variable $z$ serves as an invertible representation of the input CT, enabling the reconstruction of a distribution of plausible harmonized images rather than a single deterministic output.

We trained CTFlow under two schemes: mapping Medium/25% images to reference images (i.e., CTFlow-Base) and to texture-transformed reference images (i.e., Cycle-CTFlow). The impact of harmonization was evaluated on two datasets. The first is the publicly available MiniDeepLesion dataset from Kaggle, which contains 88 lung nodule cases derived from the NIH DeepLesion dataset (Yan et al., 2018). The second dataset comprises our in-house collection of diagnostic dose CT scans from never-smokers ($< 100$ lifetime cigarettes) and ever-smokers, referred to as the UCLA in-house dataset, and includes 112 lung nodule cases.

## 3. Results and Discussion

Table 1 shows that Cycle-CTFlow consistently outperforms the baseline model across both datasets, achieving the highest sensitivity and CPM scores, while yielding fewer false negatives, which is crucial in the context of lung cancer detection. It should also be noted that Cycle-CTFlow results in more false positives. We demonstrate that the texture-transformed training scheme is a more effective approach to harmonization, and our future efforts will focus on improving this module, extending beyond the current reliance on CycleGAN.

## Acknowledgments

The authors acknowledge funding for this work from the NIH/National Institute for Biomedical Imaging and Bioengineering under award number R01 EB031993. The content is solely the responsibility of the authors and does not necessarily represent the official views of the sponsoring agency.

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
