# OpenReview forum: "Cycle-CTFlow: A CycleGAN–Normalizing Flow Harmonization Framework for Improved Nodule Detection"
_MIDL.io/2025/Short_Papers — MIDL 2025 - Short Papers_

### Official Review · Reviewer_2LEC · 2025-04-28

**Rating:** 4
**Confidence:** 4

**Summary:**

This work proposes using a CycleGAN combined with CBAM for domain adaptation across CT datasets. Experiments on two lung nodule detection datasets demonstrate the effectiveness of the proposed approach.

**Strengths:**

•	The problem addressed is practically significant, and the proposed methodology is reasonable.

**Weaknesses:**

•	Since in-house samples are used during training, the validation results on the same in-house dataset do not fully reflect the domain adaptation capability of the method.
•	There are several formatting issues: for example, Figure 1 overflows beyond the left page margin, and Table 1 is not centered.

---

### Decision · Program_Chairs · 2025-05-01

Accept